# Tumor Cell-Intrinsic Immunometabolism and Precision Nutrition in Cancer Immunotherapy

**DOI:** 10.3390/cancers12071757

**Published:** 2020-07-02

**Authors:** Elisabet Cuyàs, Sara Verdura, Begoña Martin-Castillo, Tomás Alarcón, Ruth Lupu, Joaquim Bosch-Barrera, Javier A. Menendez

**Affiliations:** 1Program Against Cancer Therapeutic Resistance (ProCURE), Metabolism and Cancer Group, Catalan Institute of Oncology, 17007 Girona, Spain; ecuyas@idibgi.org (E.C.); sverdura@idibgi.org (S.V.); 2Girona Biomedical Research Institute, 17190 Salt, Girona, Spain; bmartin@iconcologia.net (B.M.-C.); jbosch@iconcologia.net (J.B.-B.); 3Unit of Clinical Research, Catalan Institute of Oncology, 17007 Girona, Spain; 4ICREA (Institució Catalana de Recerca i Estudis Avançats), 08010 Barcelona, Spain; talarcon@crm.cat; 5Centre de Recerca Matemàtica (CRM), 08193 Bellaterra, Barcelona, Spain; 6Departament de Matemàtiques, Universitat Autònoma de Barcelona, 08193 Bellaterra, Barcelona, Spain; 7Barcelona Graduate School of Mathematics (BGSMath), 08193 Bellaterra, Barcelona, Spain; 8Department of Medicine and Experimental Pathology, Mayo Clinic, Rochester, MN 55905, USA; lupu.ruth@mayo.edu; 9Mayo Clinic Cancer Center, Rochester, MN 55905, USA; 10Medical Oncology, Catalan Institute of Oncology, Dr. Josep Trueta Hospital of Girona, 17007 Girona, Spain; 11Department of Medical Sciences, Medical School University of Girona, 17003 Girona, Spain

**Keywords:** immune checkpoints, immune checkpoint inhibitors, metabolism, nutrition, diet

## Abstract

One of the greatest challenges in the cancer immunotherapy field is the need to biologically rationalize and broaden the clinical utility of immune checkpoint inhibitors (ICIs). The balance between metabolism and immune response has critical implications for overcoming the major weaknesses of ICIs, including their lack of universality and durability. The last decade has seen tremendous advances in understanding how the immune system’s ability to kill tumor cells requires the conspicuous metabolic specialization of T-cells. We have learned that cancer cell-associated metabolic activities trigger shifts in the abundance of some metabolites with immunosuppressory roles in the tumor microenvironment. Yet very little is known about the tumor cell-intrinsic metabolic traits that control the immune checkpoint contexture in cancer cells. Likewise, we lack a comprehensive understanding of how systemic metabolic perturbations in response to dietary interventions can reprogram the immune checkpoint landscape of tumor cells. We here review state-of-the-art molecular- and functional-level interrogation approaches to uncover how cell-autonomous metabolic traits and diet-mediated changes in nutrient availability and utilization might delineate new cancer cell-intrinsic metabolic dependencies of tumor immunogenicity. We propose that clinical monitoring and in-depth molecular evaluation of the cancer cell-intrinsic metabolic traits involved in primary, adaptive, and acquired resistance to cancer immunotherapy can provide the basis for improvements in therapeutic responses to ICIs. Overall, these approaches might guide the use of metabolic therapeutics and dietary approaches as novel strategies to broaden the spectrum of cancer patients and indications that can be effectively treated with ICI-based cancer immunotherapy.

## 1. Introduction

In 2011, Hanahan and Weinberg added the evasion of immune destruction as a new trait to the list of cancer hallmarks [1]. Since then, the use of inhibitors targeting the specific mechanisms that cancer cells employ to evade immune system detection, particularly the programmed cell death protein-1 (PD-1)/programmed cell death-ligand 1 (PD-L1), and cytotoxic T lymphocyte-associated protein-4 (CTLA-4) immune checkpoint axes, has revolutionized the therapeutic landscape of tumors previously considered to be highly lethal [2,3,4,5]. Despite the unquestionable success of anti-PD-1 (pembrolizumab and nivolumab), anti-PD-L1 (atezolizumab, durvalamab, and avelumab), and anti-CTLA-4 (ipilimumab) monoclonal antibodies for extending overall survival over standard chemotherapy and genomically-targeted therapies in an ever-growing list of malignancies, the clinical gains of these so-called immune checkpoint inhibitors (ICIs) are far from universal. Indeed, this ground-breaking therapeutic modality does not benefit a majority (>80%) of cancer patients [6,7]. The ubiquity of innate (primary) and adaptive resistance prevents the introduction of ICIs into the therapeutic armamentarium for some of the major cancer types (e.g., breast, prostate, pancreas). One third of patients with acquired (secondary) resistance relapse after initial response, pointing to the coexistence of multiple, non-redundant tumor cell-intrinsic and -extrinsic mechanisms of cancer immunotherapy resistance. The biggest challenge in the cancer immunotherapy field is now the need to biologically rationalize and broaden the clinical utility of ICIs in a majority of unresponsive cancer patients across multiple tumor types.

Much effort has been invested into the identification of response markers and the mechanisms of tumor response to cancer immunotherapy, such as the expression of immune checkpoint proteins, tumor mutational burden, and immune/inflammation gene signatures [6,7,8,9,10]. Although the predictive value of these histological, genomic, and transcriptomic approaches remains limited, they are beginning to provide key mechanistic insights into the clinical benefit of potentiating tumor T-cell infiltration for boosting responses to ICIs via the incorporation of, for example, anti-cytokine (IL-1β, IL-6, TGFβ) immunotherapeutics [11,12,13,14,15]. Intriguingly, there is a solid body of evidence suggesting that immune evasion and metabolic reprogramming [16,17,18,19], which also attained the status of emerging cancer hallmark in 2011 [1], are fundamentally linked. Indeed, the metabolic instruction of host immunity and cancer immune evasion has crucial implications for the use of cancer metabolic traits to circumvent some of the current weaknesses of cancer immunotherapy including its (lack of) universality and durability.

The past decade has witnessed the emergence of cancer immunometabolism as a flourishing research area at the interface between the historically distinct disciplines of immunology and metabolism [20,21,22]. Cancer immunometabolism continues to gain momentum through the realization that metabolic remodeling underlies cancer immune responses, and that targeting immune and cancer cells’ metabolism can bolster antitumor immunity and synergize with cancer immunotherapy. There has been much focus on understanding how the immune system’s ability to kill tumor cells requires a conspicuous metabolic specialization of T-cells (Figure 1, layer 1). We are also beginning to appreciate how cancer cell-associated metabolic activities trigger shifts in the abundance of certain metabolites with immuno-regulatory roles in the tumor microenvironment (TME) (Figure 1, layer 2). Correspondingly, cancer metabolism research has entered a long-awaited therapeutic era in which a novel category of immunometabolic drugs are being clinically tested to boost the metabolic demands of immune cells and suppress the immunosuppressive metabolic features of the TME [23,24]. That being said, very little is known about the specific tumor cell-intrinsic metabolic traits that control the immune checkpoint contexture in cancer cells (Figure 1, layer 3). Likewise, we are lacking a comprehensive understanding of how systemic metabolic perturbations in response to certain dietary approaches can reprogram the immune checkpoint landscape of tumor cells.

In this perspective article, we first review how efforts made to better understand the metabolic nature of the immunosuppressive TME has already led to various combinations of immunotherapies with cancer metabolism-targeting drugs. We then present the notion that a comprehensive appreciation of how cancer cell-intrinsic metabolic changes and -extrinsic dietary influences regulate essential immune checkpoints for T-cell activation (and/or alternative mechanisms of immune escape) is urgently needed to successfully target the complex immuno-metabolic interplays occurring between different cellular compartments of the TME. We finally propose that a molecular-level understanding and functional immunomonitoring of how cancer cell-autonomous metabolic traits and diet-mediated changes in nutrient availability and utilization impact the immune checkpoint landscape might guide the incorporation of anti-metabolic drugs and dietary approaches into frontline treatments with ICI-based immunotherapy. While we acknowledge that immune checkpoint molecules expressed on natural killer (NK) and tumor-infiltrating myeloid cells have recently emerged as promising therapeutic targets in cancer immunotherapy [25,26,27,28,29,30], the influence of NKs and myeloid cell metabolism on their immune checkpoint composition and on ICI efficacy is not considered in detail here.

## 2. Metabolic Functions of Immune Checkpoints and the Metabolic Nature of Immunosuppressive TME: Battling over Metabolic Resources

The ability of tumors to impose desert/exclusion patterns in the topography of cancer-associated immune cells largely relies on the promotion of metabolic layers of defense (Figure 1). Such defensive regulation closely relates to the immunometabolic regulatory actions of immune checkpoints on T-cells and the metabolic specialization that is required to drive the differentiation, function, and trafficking of T-cells into tumor sites [31,32,33,34].

The capacity of immune checkpoints to operate as potent inhibitors of anti-tumor T-cell responses relies, at least in part, on their ability to impede the utilization of nutrients such as glucose and glutamine and some specific species of fatty acids that, altogether, allow the activation, phenotypic switching, and self-renewal of T-cells. Engagement of PD-1 with PD-L1 expressed on the tumor cell surface drives a condition called immune metabolic anergy that prevents T-cells from meeting the metabolic demands required to sustain anti-tumor responses [35,36]. Indeed, a restoration of the metabolic preferences and mitochondrial fitness of tumor-infiltrating T-cells is recognized as a pivotal mode of action of ICIs. The paradoxical association between obesity and increased anti-tumor efficacy and the clinical outcomes of ICIs serves to illustrate how dynamic reprogramming of metabolism is essential for T-cell functioning [37,38]. Obesity increases the metabolic activity and growth of tumors and promotes immune dysfunction, in particular by the upregulation of PD-1, which in turn augments the energy metabolism and mitochondrial biogenesis of T-cells. Obesity-driven constant changes in the metabolic and activation status of T-cells could lead to a terminally differentiated exhausted T-cell phenotype. This phenomenon amplifies the biological effects and clinical benefits that arise from the metabolic reinvigoration of T-cells driven by anti PD-1/PD-L1 therapeutics.

Unrestrained tumor growth leads to the excessive consumption of the same set of nutrients that are essential for effector T-cells, including glucose, amino acids, and fatty acids. Effector T-cells entering the metabolically inhospitable TME face stressful constraints on the metabolic fitness required for their optimal proliferation and anti-tumor functionality [37,39,40,41,42,43,44,45,46]. Such nutritional competition phenomena are further amplified by the tumor-driven recruitment of immune suppressor cells such as regulatory T-cells (T_reg_), M2 tumor-associated macrophages (TAMs), and myeloid-derived suppressor cells (MDSCs). This immunosuppressive cell compartment depends on specific metabolic functions to exert its functions, or can promote high expression of metabolic enzymes in dendritic cells, such as arginase and indolamine 2,3-dioxygenase (IDO), thereby reducing the availability of key amino acids required by T-cells such as arginine and tryptophan [37,46]. Kynurenine, the catabolic product of IDO-driven tryptophan metabolization in tumor cells and TAMs, can further impede T-cell activation and promote the development of immunosuppressive T_reg_ [24]. Extracellular adenosine, which is generated by the elevation of the ectonucleotidases CD39 and CD73 in the TME [47,48], is a potent metabolic immunosuppressor that directly restricts antitumor responses by activating the adenosine A_2A_ receptor (A_2A_R) in T-cells. The TME can acquire additional immunosuppressive traits via tumor cell-driven release of immune regulatory metabolic (by)products that can blunt T-cell activity while promoting the differentiation of immunosuppressive cell subsets [39,40,41,42,43,44,45]. This is clearly illustrated by the metabolic product lactate and the oncometabolite (R)-2-hydroxyglutarate (R-2HG), both of them operating as bona fide immunosuppressive metabolites by promoting the proliferation of immunosuppressive cell subsets while significantly limiting the tumor bed recruitment, proliferation, and functionality of T-cells [49,50,51,52].

### 2.1. Metabolic Drugs Targeting the Immunosuppressive Nature of TME: A First-in-Class Success of the Cancer Metabolism Pipeline

The ability of ICIs to relieve the inhibitory signals of immune checkpoints towards the metabolically active immuno-phenotype of T-cells is expected to synergize with other therapeutic approaches aimed at suppressing the metabolic quenching activity of the TME. Indeed, although the innate flexibility of cancer cell metabolic pathways has been a thorn in the side of cancer metabolism drug developers, a remarkable exception is the rise of a novel category of metabolic drugs aimed to overcome immunotherapy resistance by improving the metabolic competitiveness and functionality of T-cells in the TME [23,24]. Blocking glucose metabolism with mammalian target of rapamycin (mTOR) and hexokinase II inhibitors, oxidative phosphorylation with inhibitors of the mitochondrial electron transport chain, lactate transport with monocarboxylate-transporter inhibitors, lipid metabolism with cyclooxygenase (COX) inhibitors, adenosine metabolism with anti-ectonucleotidase antibodies, and preventing tryptophan, arginine, and glutamine depletion with IDO-, arginase-, and glutaminase-inhibitors, are all examples of the numerous preclinical approaches that have been proposed to rewire the metabolic fitness of T-cells in the TME [24,53,54,55]. Numerous phase I/II trials are currently underway combining inhibitors of the glutamine/glutamate pathway, arginine pathway, 2HG-producing mutant isocitrate dehydrogenase (IDH), the adenosine pathway, cyclooxygenase 2 (COX2) and/or prostaglandin E2 (PGE_2_), and glucose metabolism, with a wide variety of ICI partners [23,24]. Of note, first-in-human clinical trials of agents targeting the adenosinergic signaling pathway as single agents and in combination with anti-PD-1/PD-L1 therapies are demonstrating high rates of disease control [24,47,48]. On the negative side, a phase III study combining an IDO1-selective competitive inhibitor with an anti-PD-1 monoclonal antibody failed to provide significant therapeutic gain in patients with metastatic melanoma [56,57,58,59]. These clinical findings highlight the need for uncovering the metabolic flexibility dependencies of tumor cells in responding or adapting to metabolic modulators [60,61,62,63,64,65], including dietary approaches [65,66,67], aimed at overcoming resistance to ICIs.

#### 2.1.1. Phosphatidylinositol 3-Kinase (PI3K) Inhibitors and Insulin Signaling

Despite clear molecular evidence of the PI3K cascade as one of the most frequently activated metabolic drivers of cancer, small-molecule PI3K inhibitors have yielded disappointing results in monotherapy trials [68,69,70]. Pan-PI3K inhibitors induce transient hyperglycemia and, consequently, a compensatory increase in insulin production and secretion that can reactivate PI3K signaling in tumor cells [70]. This metabolic nullification of the expected beneficial effects arising from the therapeutic blockade of PI3K could be successfully overcome by limiting circulating glucose and insulin/insulin-like growth factor-I (IGF-I) levels. Accordingly, a ketogenic (i.e., high-fat, very low-carbohydrate) diet preventing hyperglycemia and lowering insulin release has been shown to efficiently reduce the activation of insulin receptors in tumors and to restrict the re-activation of the PI3K pathway in tumor-bearing mice, thereby enhancing the therapeutic response to PI3K inhibitors [18]. While it might be argued that this approach exclusively holds promise for patients with tumors bearing mutations in PIK3C genes being treated with PI3K inhibitors, the implications are wider and motivate its consideration in ICI-based cancer immunotherapy. For instance, a ketogenic diet has been associated with a reduction in the (PI3K-driven) expression of PD-L1 and an increase in tumor-infiltrating CD8^+^ T-cells [44,45,71], thus highlighting how systemic metabolic perturbations in response to dietary interventions can reprogram the immune checkpoint landscape of tumor cells for anticancer immune response activation (Figure 2, left). 

#### 2.1.2. IDO Inhibitors and the Trypophan-Kynurenine Pathway

The IDO pathway mediates immunosuppressive effects through the metabolization of tryptophan to kynurenine [24,46,58]. Two main enzymes, IDO and tryptophan-2,3-dioxygenase (TDO), regulate the first and rate-limiting step of the kynurenine pathway (Figure 2, right). IDO operates as an immune checkpoint in peripheral immune tolerance through its capacity to inhibit the proliferation of T-cells and sensitize them to apoptosis via the starvation of tryptophan. In addition, other cells that reside in the TME (e.g., dendritic cells and MDSCs) often express IDO and help tumor cells to accumulate kynurenine, which signals through the aryl hydrocarbon receptor (AhR) to promote tumor tolerance to tumor-infiltrating cytotoxic T-cells. While the combination of IDO inhibition and blockade of the PD-1/PD-L1 axis has a strong rationale, as these immune checkpoints inhibit immune response via complementary mechanisms, the ECHO-301/Keynote-252 trial failed to demonstrate a clinical benefit of adding the IDO inhibitor epacadostat to the anti-PD-1 monoclonal antibody pembrolizumab in patients with unresectable metastatic melanoma [57]. The inhibition of IDO could trigger the induction of the co-expressed TDO as a compensatory immunosuppressive mechanism and, therefore, IDO inhibition alone will likely not suffice to prevent kynurenine-mediated immunosuppression in most cancers. The sequential or simultaneous inhibition of both IDO and TDO with IDO-specific and TDO-specific or dual IDO/TDO inhibitors might be a preferable course of treatment to relieve the immunosuppressive effect of kynurenine. Inhibitors of IDO/TDO downstream mediators where the pathway converges (i.e., enzymes such as kynurerinase and small-molecule antagonists of AhR) would be an alternate way of repressing kynurenine-mediated immunosuppression. Moreover, if suppression of kynurenine is indeed the dominant mechanism, the ability of the ketogenic diet to target its metabolism [72,73], may help IDO/TDO inhibitors to drive intratumoral levels of kynurenine sufficiently low to alleviate the immunosuppressive effects of kynurenine-activated AhR (Figure 2, right).

These two examples serve to highlight that either pharmacological or dietary targeting of metabolic compensatory mechanisms can improve the therapeutic outcomes of ICIs. The dependency of immunotherapy-resistant states on certain metabolic traits might rationally determine the ability of specific metabolic and/or dietary interventions to augment the tumoricidal effects of existing cancer immunotherapies. Elucidating context-specific metabolic dependences and their connections to immune checkpoints and immune-evasion programs in tumor cells represents a promising approach to identifying new metabolic and dietary approaches for ICI-based cancer immunotherapy.

### 2.2. Dietary Interventions/Modifications Impacting the Immunosuppressive Traits of TME

Diet-mediated changes in whole-body metabolism can locally alter access to and utilization of nutrients by cancer cells, which could be exploited to bolster not only the tolerability but also the efficacy of existing cancer therapies. Although originally assumed not to be relevant, certain dietary interventions/modifications have been proposed as an integral part of anti-cancer regimens—as an extrinsic metabolic means capable of impacting cancer patient outcomes [17,18,19]. The mechanistic ability of starvation/calorie restriction (e.g., fasting/intermittent fasting or fasting-mimicking diets leading to glucose restriction) and more targeted dietary approaches involving selective nutrient limitation (e.g., deprivation of amino acids such as methionine or serine) or selective nutrient supplementation (e.g., histidine, mannose) to enhance cancer therapy has been described in two recent excellent reviews [66,67]. In the context of cancer immunotherapy, we are beginning to appreciate how certain dietary interventions/modifications may alleviate the severe immunosuppressive metabolic traits of the TME. Fasting-mimicking diets appear to target the link between the A_2A_R-related enzyme heme oxygenase (HO-1) in T-cells and the production of immunosuppressive adenosine by cancer cells, thereby inducing chemosensitization phenomena and tumor growth inhibition via the enhancement of T-cell killing [74]. Ketogenic diets might overcome several immune-escape mechanisms by lowering the expression of the immune checkpoints CTLA-4 and PD-1 on TILs and increasing cytokine production and T-cell killing activity [71]. Similarly, low-protein diets or specific restriction of methionine have been shown to improve the tumor-reactive response of T-cells and macrophages [75,76].

Certain amino acids that are depleted in the interstitial fluid of tumors and in the circulation of cancer patients are required for an effective immune response [77,78]. Therefore, it might be argued that dietary approaches further limiting the availability of these amino acids to prevent the growth of cancer cells may also deprive immune cells of essential nutrients, thus having the detrimental effect of suppressing anti-cancer immunity and hindering the anti-cancer activity of ICIs. Alanine deprivation leads to metabolic and functional impairment during T-cell activation and memory T-cell restimulation [79]. Likewise, diet-induced reduction of circulating serine, which supplies glycine and one-carbon units for de novo nucleotide biosynthesis in proliferating T-cells, severely limits T-cell expansion even in the presence of optimal glucose concentrations [80]. Whereas arginine-depleted diets and arginine-degrading agents such as the PEGylated arginine deiminase (ADI-PEG20) synergize with the anti-tumor effects of cancer therapies in different mouse tumor models [67,81,82], dietary supplementation of arginine has been shown to limit the growth of cancers by enhancing innate and adaptive immune responses [83,84]. Indeed, arginine depletion using a PEGylated form of the catabolic enzyme arginase I (peg-Arg I) enhances tumor growth in mice by inducing MDSCs and blunting T-cell responses [85], confirming that arginine can have a direct impact on the metabolic fitness and survival capacity of T-cells crucial for anti-tumor responses.

While the abovementioned studies might raise concerns that the dietary restriction of certain amino acids decreases the anti-cancer response to ICIs, it should be noted that the anti-tumor efficacy of a serine-free diet has been demonstrated not only in T-cell-deficient (nude) mice but also in immunocompetent mice [17]. These considerations highlight the need for employing immunocompetent models in preclinical studies of metabolic therapies and/or dietary modifications for enhanced cancer immunotherapy. It should be underlined, however, that while mice are the experimental tools of choice for the majority of immunologists, there are notable discrepancies between mice and humans not only in their innate and adaptive immunity (reviewed in [86,87,88,89]), but also in the metabolism of immune cell populations (e.g., arginine metabolism and arginase expression in macrophages and myeloid cells [90,91]). Because these “of mice and men” variables impact both immunity and metabolism, they might also influence outcomes during immunotherapy and should be carefully considered during the optimization of humanized animal models for the translation of immuno-metabolic strategies to human clinical trials.

The composition of the gut microbiota is known to impact the efficacy of ICIs and changes in host metabolism and microbiota can occur in tandem [92,93,94]. Dietary modifications could therefore alter not only the circulating composition of immunologically relevant metabolites (e.g., glucose, ketone bodies) and immunological factors (e.g., inflammatory cytokines), but also gut microbiota assets positively related to an improved efficacy of ICIs—*Bifidobacterium* and *Faecalibacterium,* among others [95]. Accordingly, the ketogenic diet has been shown to increase the relative gut microbiota abundance of *Akkermansia muciniphila* [96], a bacterium capable of restoring the response to immune checkpoint blockade in cancer models [97]. Reduction in dietary methionine/cysteine intake might increase immunotherapy efficacy, at least in part, via changes in gut microbiota [76,98]. Although unlikely, it cannot be excluded that the ability of microbiota to synthesize specific nutrients (e.g., amino acids, short-chain fatty acids) for the host might potentially circumvent the immunological effectiveness of dietary interventions/modifications [99].

## 3. Cell-Intrinsic Metabolic Traits and The Immune Checkpoint Composition of Tumor Cells: A Forgotten Dimension of Cancer Immunometabolism

There has been a paucity of studies examining how tumor cell-intrinsic and -extrinsic (e.g., dietary) determinants of the metabolic features of cancer cells might alter their immune evasion strategies, including the composition of the immune checkpoint landscape. The recently uncovered association between mitochondrial metabolism and the antigen presentation machinery of tumor cells has illuminated a largely unexplored dimension of cancer immunometabolism—namely the dependence of tumor immunogenicity and immunotherapy responsiveness on the metabolic state of tumor cells [100,101,102,103].

The most common metabolic changes occurring in cancer cells are closely intertwined with aberrations in oncogenic and tumor-suppressive pathways that are known to contribute to the expression status of immune checkpoints such as PD-L1 (e.g., Phosphatase and tensin homolog (PTEN)/liver kinase B (LKB) deletions, PI3K/protein kinase B (AKT) mutations, MYC overexpression, signal transducer and activator of transcription 3 (STAT3) activation, etc.) [104,105,106,107]. Dysregulated activation of immune checkpoints might therefore be viewed as a general cancer cell-autonomous mechanism of metabolism-driven tumor immune-tolerance. Oncogenic activation of the archetypal PI3K-AKT-mTOR metabolic pathway, which coordinates the uptake and utilization of multiple nutrients including glucose, glutamine, nucleotide, and lipids, promotes immune escape by driving PD-L1 overexpression in tumor cells. The fact that PD-L1 protects cancer cells from immune-mediated cell death via activation of the PI3K/AKT pathway and mTOR [108] supports the notion that dysregulated cancer cell-autonomous metabolism might represent a two-way barrier against antitumor immunity. Also, pyruvate kinase muscle 2 (PKM2), the alternative splicing form of PKM that enables exacerbated aerobic glycolysis in cancer cells, has been shown to directly promote the expression of PD-L1 in cancer cells [109,110]. Further, select metabolic activities and metabolites might enable cancer cells to simultaneously drive immunologically relevant decisions on both immune and tumor cell compartments. Tumor cell-derived oncometabolites such as R-2HG can be taken up by T-cells to inhibit histone and DNA methylation, perturbing the epitranscriptional programs of T-cells and ultimately resulting in suppressed T-cell proliferation and effector functions [52,111]. We and others have recently shown that this very same ability of the oncometabolite R-2HG to influence chromatin functioning also epigenetically alters the expression of *PD-L1* in cancer cells themselves [112,113] (Figure 3). Accordingly, the so-called “immunologically quiet” immune-cancer subtype, which is highly enriched in tumor types bearing R-2HG-producing mutations in the metabolic enzyme IDH [41], is characterized by fewer tumor-associated immune cells.

From a therapeutic perspective, tumor cell aerobic glycolysis associates with a reduced expression of major histocompatibility complex (MHC) proteins [114,115], which are known to confer differential sensitivity to CTLA-4 and PD-1 blockade [116]. Because MHC recovery approaches are expected to synergize with complementary forms of immunotherapy [117], it is noteworthy that enhancement of mitochondrial biogenesis with the mitochondrial complex I inhibitor metformin suffices to promote MHC-I expression in cancer cells [118]. The potential synergy between immunotherapies and the metabolic ability of metformin to restore antigenicity and immunogenicity in tumor cells is currently under investigation in a number of clinical trials [24,119]. Other cancer cell-autonomous metabolic traits shaping tumor cell-intrinsic immune evasion mechanisms that may play a key role in immunotherapy response are yet to be discovered.

Combinations of dietary approaches including fasting, low-calorie fasting-mimicking diets, and (high-fat low-protein/carbohydrate) ketogenic diets with chemotherapy, immunotherapy or other cancer treatments might be promising strategies to reduce treatment-related adverse effects and boost efficacy outcomes [19]. The ability of such dietary strategies that are capable of enhancing the potency and durability of tumor-specific T-cells [120,121] to directly regulate the expression of immune checkpoint and improve immune response in conjunction with anti-PD-1/PD-L1 drugs is largely unexplored. Remarkably, we recently observed that mimicking caloric restriction- and ketogenic diet-like milieus by decreasing available glucose and increasing blood ketones such as β-hydroxybutyrate downregulated the content of cell membrane-associated PD-L1 in an experimental model of highly aggressive basal-like breast cancer [122] (Figure 4). These findings, although preliminary, support the notion that diet-mediated changes in nutrient availability/utilization could impact the immune checkpoint composition of cancer cells and, as a consequence, the responsiveness to ICIs.

We are coming to appreciate the impact of tumor cell-intrinsic metabolism and the importance of nutrient supply in determining immune surveillance and ICI therapy outcomes. Beyond the aforementioned examples, however, the appraisal of the intrinsic metabolic dysregulation of cancer cells as a bona fide driver of both constitutive and inflammation-inducible expression of immune checkpoints’ composition and its association with the response to immunotherapy is a largely neglected aspect of cancer immunometabolism. Such a limitation, which may also involve unforeseen metabolic dependencies of tumor cell-intrinsic immune evasion mechanisms irrespective of the status of immune checkpoints, may preclude the expected therapeutic potential of ongoing combination therapies with metabolic agents and immunotherapies given the similar metabolic needs of cancer and immune cells. We are also missing a detailed understanding of how diet-mediated extrinsic changes in whole body metabolism and systemic nutrient availability can impact the immune checkpoint composition of tumor cells, with potentially important consequences for the outcomes of currently existing ICIs. Indeed, a rational combination of dietary approaches and immunotherapy will need a detailed understanding of the genetic and non-genetic mechanisms that determine the metabolic dependencies of the immune checkpoint contexture in tumor cells. Such a limitation might prevent scientific examination aimed to unambiguously probe how the connections between diet and cancer metabolism could be exploited to bolster the clinical efficacy of cancer immunotherapy. To circumvent such boundaries, we need to take advantage of wide exploratory frameworks to examine how cancer cell-metabolic state and dietary interventions drive cancer cell-intrinsic immune-evasion mechanisms—including the composition of the immune checkpoint landscape—and consequently dictate the responsiveness of tumor cells to cancer immunotherapies (Figure 5).

### 3.1. How Can We Examine the Cancer Cell-Autonomous Metabolic Dependencies of the Immune Checkpoint Landscape?

The regulatory mechanisms of immune checkpoint (e.g., PD-L1) expression and activity in cancer cells, which extend from gene amplification, chromatin modification, and transcription to translation and post-translational modification (PTM) [123,124,125,126], are all amenable to metabolic regulation (Figure 6).

First, the ability of oncometabolite prototypes such as 2HG to epigenetically regulate PD-L1 expression exemplifies the likely existence of active metabolites capable of providing direct conduits that connect the aberrant functioning of metabolic enzymes to the epigenetic regulation of immune checkpoint genes in cancer cells. An increased generation of oncometabolites serving as epigenetic modifiers is a key feature of cancer subtypes and cancer cell states enriched with characteristics of the epithelial-to-mesenchymal transition (EMT) program [127,128,129,130,131,132,133,134], one of the major contributors to primary and acquired resistance to PD-L1-targeted immunotherapy [135,136,137,138]. Such metabolo-epigenetic regulatory mechanisms might be of relevance to molecularly understand and clinically capture the heterogeneous and dynamic nature of PD-L1 expression within tumors and between different tumor sites, which largely limits the clinical value of measuring PD-L1 expression status as a companion diagnostic for the benefit-risk assessment of PD-1/PD-L1 inhibitors [139,140]. Beyond the potential oncometabolites that can result from loss-/gain-of-function mutations in metabolic enzymes, major metabolic pathways controlling the abundance of numerous “common” metabolites that act as co-factors and co-substrates of chromatin-modifying enzymes could also influence the epigenome structure and function to alter the expression of PD-L1 in cancer cells. Second, PTMs of PD-L1 such as glycosylation, phosphorylation, ubiquitination, sumoylation, and acetylation play key roles in the regulation of PD-L1 protein stability, translocation, and protein–protein interactions, suggesting the likely existence of metabolic chemical modifications of immune checkpoints. Because the abundance of many of these metabolite-induced PTMs is directly dependent on the metabolic state of cancer cells [141], such a powerful means of immune-phenotype modulation should be carefully considered when evaluating the impact of cancer cell metabolism on the functionality of the immune checkpoint landscape. Accordingly, we and others have shown that the AMP-activated protein kinase (AMPK)-sensed energetic crisis imposed by the small-metabolic molecule metformin reduces the stability and membrane localization of PD-L1 by inducing its endoplasmic reticulum (ER)-associated protein degradation (ERAD) in cancer cells [119,142]. In response to metformin, the activated form of AMPK directly phosphorylates PD-L1 in a manner that promotes its abnormal glycosylation, resulting in ER accumulation and ERAD, which contributes to enhanced cytotoxic T-cell activity against metformin-treated cancer cells. We recently documented how the dietary polyphenol resveratrol significantly reduces the immune evasion activity of PD-L1 by operating as a direct inhibitor of the glyco-PD-L1-processing enzymes (α-glucosidase/α-mannosidase) that modulate *N*-linked glycan decoration of PD-L1, thereby promoting the ER retention of a mannose-rich, abnormally glycosylated form of PD-L1 and, consequently, increasing the susceptibility of cancer cells to T-cell-mediated cell death [143]. Third, a completely unexplored layer of complexity in the immune-metabolic network might involve the non-covalent modification of immune checkpoints via metabolite-driven assembly of higher molecular proteins [141] and metabolite-controlled transcription and translation performed by riboswitches [144,145]. Because metabolites that control riboswitches include lysine, glutamine, cobalamin, thiamine, pyrophosphate, purines, and S-adenosylmethionine (SAM), this regulatory mechanism, which can also extend to transcription factors, might provide a new means through which the active metabolome determines the response of cancer cells to immune-regulatory cues via the control of immune checkpoint expression.

Research designs involving metabolic gene-focused knockout and transcriptional activation screening using the clustered regularly interspaced short palindromic repeats (CRISPR)/CRISPR-associated protein 9 (Cas9) system can be extremely helpful for deciphering tumor-intrinsic metabolic traits capable of regulating the expression and/or functionality of PD-L1 (and other immune checkpoints) in cancer cells. CRISPR/Cas9-based functional genomic screens have been shown to provide a high-throughput means of functionally characterizing the essentiality and context-dependent dispensability of central metabolic pathways for cancer cell growth under certain metabolic conditions [146,147,148]. As such, they hold promise also for defining the cancer cell-autonomous metabolic requirements not only of the expression of immune checkpoints, but also of unforeseen metabolic dependencies of tumor cell-intrinsic immune evasion mechanisms irrespective of the immune checkpoint landscape status. Because the plasticity and redundancy of metabolic networks might allow these experimental systems to remodel around a single knock-out, combinatorial CRISPR metabolic screens simultaneously targeting different metabolic genes could be employed to systematically map the dispensability and interactions of whole groups of metabolic genes (e.g., those encompassing glycolysis, gluconeogenesis, the pentose phosphate pathway, tricarboxylic acid cycle (TCA) cycle, etc.) to regulate the expression of immune checkpoints in cancer cells. Improving the metabolic fidelity of in vitro cancer models with recently developed culture media better representing physiological nutritional features [149,150] could make the CRISPR-based screening of “metabolo-immune checkpoints” highly informative and physiologically relevant.

### 3.2. How Can We Examine the Impact of Dietary Interventions on the Efficacy of Cancer Immunotherapy?

A better understanding of the interactions between the metabolic state of cancer cells and the systemic macro- and local micro-environmental nutritional conditions is critical for considering how dietary interventions might impact the metabolic dependences of the immune checkpoint landscape and tumor responsiveness to ICIs.

First, diet-induced changes in nutrient availability (and utilization) within the TME may perturb specific metabolic pathways that operate as drivers of immune checkpoints. Because the metabolic requirements of commonly employed cell culture models and tumors in vivo are very different, a disproportionate nutrient composition of commercial media might impose undesirable metabolic artifacts when assessing the impact of dietary interventions in the efficacy of ICIs in vitro. Although our capacity to successfully overcome this limitation is still extremely limited [151], functional screenings to characterize the impact of dietary interventions on the efficacy of ICIs would be performed using more physiological tissue culture conditions that recapitulate some metabolic features of the tumor interstitial fluid [149,150,152,153], and simultaneously modifying those metabolites that are known to be affected by specific diets. These so-called metabolite add-back or dropout experiments can be coupled to transcriptomic and metabolomics/fluxomics platforms to identify the specific metabolic perturbations responsible for changes in immune-escape mechanisms including the composition of the immune checkpoint landscape. The targeting of these pathways with diet mimetics (e.g., rapalogs, biguanides, sirtuin-activating compounds, and nicotinamide adenine dinucleotide (NAD^+^) precursors) could also be explored to test their ability to drive the same immune-modulatory effects, and to clinically replace non-appealing and/or difficult-to-adhere-to dietary interventions such as calorie restriction. Second, the identification of metabolic pathways conferring resistance either to the immune-regulatory effects of a given dietary modification and/or to cancer immunotherapy might provide evidence for synthetic lethal metabolic vulnerabilities that can be therapeutically exploited in a clinical setting. This maximal utility of dietary interventions based on the co-dependency of immunotherapy-resistant states on certain metabolic traits might rationally determine whether certain diets will synergize with or antagonize the tumoricidal effects of existing ICIs. Once the metabolic determinants of positive/negative cancer cell responsiveness have been identified (e.g., by using metabolic gene-focused CRISPR-based screenings) [154,155,156], small molecule metabolic inhibitors targeting glycolysis, glutamine metabolism, lipid synthesis, or redox balance could be employed to rapidly interrogate the functional requirements of tumor cells for well-known metabolic enzymes or pathways to respond to diet-ICI combinational strategies [157,158,159]. Dietary approaches can be matched with small-molecule metabolic therapeutics pleiotropically enhancing the effects of cancer immunotherapy (e.g., the folate pathway inhibitors pemetrexed and metformin [160,161,162]) based on the metabolic vulnerabilities of tumors, thereby allowing a more customized approach to ICI-based immunotherapy.

Determining whether the system-level identification of signaling nodes essential for the metabolic and dietary regulation of immune-escape mechanisms (e.g., immune checkpoint composition) actually alter tumor cell responses to ICIs might be challenging. Over the past decade, there has been interest in adopting new strategies for reproducibly assessing the strength of the cytotoxic function of T-cells both in preclinical target evaluation and in immunomonitoring the efficacy of targeted immunotherapies. For many years, the gold standard for this purpose has been the chromium release assay (CRA) [163], an end-point assay that carries several disadvantages including the use of radioactivity, the choice of target cells, and the impossibility to assess the lysis kinetics of T-cells. Recently, novel technical solutions have permitted the dynamic detection of T-cell-mediated cytotoxicity by continuous assessment of electrical impedance using the xCELLigence real-time cell analysis (RTCA) system^®^ (Roche Applied Science, Indianapolis, IN, USA and ACEA Inc., San Diego, CA, USA) [164,165,166]. The integration of an impedance-based cytotoxic assay for the real-time and label-free assessment of T-cell-mediated killing of cancer cells with CRISPR metabolic screens would be a valuable and highly sensitive tool to enhance, in an unbiased high-throughput format, our presently limited in vitro capacity to infer the metabolic dependences of the immune-escape machinery in cancer cells and the dietary influences on the efficacy of currently existing ICIs.

## 4. Clinical and Molecular Monitoring of Tumor Cell-Intrinsic Metabolic Resistance to Cancer Immunotherapy

Little is known about the association between the metabolic state of cancer cells and the tumor response to ICIs in a clinical setting. The elucidation of tumor cell-intrinsic metabolic mechanisms of resistance to immune checkpoint blockade might illuminate actionable strategies for improving the efficacy not only of ICIs, but also of other modes of cancer immunotherapy.

There has been extensive genomic and transcriptomic research on the molecular definition and clinical discrimination of responders and non-responders to immune checkpoint blockade. A rapidly growing wave of research has (re)discovered numerous intrinsic mechanisms that can prevent a cancer patient from ever responding to immunotherapeutics including the alteration of signaling pathways (Mitogen-activated protein kinases (MAPK), PI3K, WNT, interferon (IFN)), lack of antigenic mutations, de-differentiation with loss of tumor antigen expression, alterations in antigen processing machinery, oncogenic/constitutive PD-L1 expression, and loss of human leukocyte antigens (HLA) expression. We have also identified numerous mechanisms facilitating relapse after an initial response such as the loss of T-cell functionality, loss of target antigen presentation, mutational loss of HLA, and development of escape mutations in IFN signaling [167,168,169]. Beyond the currently performed molecular analyses (e.g., mutational load, driver mutations, gene expression of immunosuppressive factors) and immune profiling (PD-L1 expression, CD8^+^ T-cells, T-cell clonality), the so-called “absence of inhibitory metabolism” (i.e., plasma lactate dehydrogenase levels, glucose utilization) is beginning to be incorporated into immunogram models to better understand the interaction between the tumor and immune cells in the TME [170]. However, there are limited studies assessing in depth the tumor cell-intrinsic metabolic dependences of primary (i.e., no active response), adaptive (i.e., initial response rapidly turned off by checkpoints or other resistance mechanisms), and acquired (i.e., response for a period of time and later progression) resistance to ICIs (Figure 7).

Unbiased systems biology approaches involving the integration of metabolomics with other omics platforms (e.g., genomics, transcriptomics, proteomics) in longitudinal tumor samples throughout the course of ICI-based treatments certainly represent the most powerful schema for identifying novel tumor cell-intrinsic metabolic mechanisms of response and resistance to immune checkpoint blockade. Alternatively, the integration of metabolomics with another omic platform at a simpler level (e.g., metabolo-transcriptomics, proteo-metabolomics) might suffice to provide an incremental discriminative/predictive capacity to currently employed histological and genomic approaches (e.g., expression of checkpoint proteins, mutational and neoepitope loads, immune gene signatures) [171,172], which can correlate with response rate but might overlap between responders and non-responders [173]. Thus, a longitudinal evaluation of metabolic changes/responses in fresh serial human specimens (e.g., tumor tissue, blood/plasma/serum, and microbiota) at pretreatment, early-treatment, and progression time points as well as a pairwise comparison of metabolic traits in pre- to in-treatment samples of responders versus non-responders [174,175], might identify dynamic metabolic biomarkers of response to ICIs as well as novel metabolic mechanisms of ICI resistance (Figure 7). With an ever-growing appreciation of the importance of monitoring treatment responses in the peripheral blood, circulating metabolites might be viewed as phenotypes serving as intermediates between genes and clinical end-points, which might facilitate the design of non-invasive liquid biopsy blood tests in the course of clinical decision making to replace conventional analyses of baseline/pretreatment metabolic markers using static unpaired biopsies.

Acknowledging the unique challenges that advanced computational analysis of multi-omic data sets per single-study subject can pose in daily clinical practice [176], such integration of metabolomic analyses into systems biology approaches hold great potential in the identification of plausible and testable metabolic networks controlling resistance to ICI-based cancer immunotherapy. Beyond the clinical correlation between the metabolic profiles in real-life biospecimens and the response to ICIs, we can functionally validate and directly associate the metabolic state of cancer cells with their immune response (Figure 7). Batteries of small anti-metabolic molecules could be employed to broadly assess the functional role of central metabolic pathways (e.g., the one-carbon (1C) cycle, glycolysis, the TCA cycle, and oxidative phosphorylation) on the immunogenicity of cancer cells. Single-gene and combinatorial CRISPR/Cas9-based metabolic screenings would be performed for the system-level identification of new metabolic nodes essential for maintaining the expression of immune checkpoints, regulating anti-tumor immunity, and dictating responsiveness to ICIs [177]. Real-time immunomonitoring of T-cell-mediated killing of cancer cells in CRISPR-edited isogenic cancer cell models engineered for single or combinational transcriptional interference and/or activation of metabolic genes in physiological-like tissue culture conditions [178,179] would be performed to rapidly infer dependencies on metabolic genes and networks of tumor cell responses to T-cells and/or ICIs. The preclinical uncovering of metabolically-driven immunosuppressive signatures might generate hypotheses for clinical validation not only in publicly available transcriptomic data from comprehensive genomic approaches [180,181], but also in longitudinal tumor samples of patients on immune checkpoint blockade (Figure 7).

## 5. Gaps and Limitations of Immunotherapy-Boosting Metabolic and Dietary Approaches

ICI-based cancer immunotherapy is associated with unpredictable immune-related adverse events (irAEs), whose mechanisms are likely to be similar to those promoting anti-tumor responses in the context of a generalized expansion of the immune response triggered by ICIs. Thus, irAEs can be mainly attributed to the exacerbation of autoimmunity (e.g., autoimmune type 1 diabetes mellitus) [182,183], but particularly to the central role of inflammation associated with checkpoint inhibitor treatments [184]. The unresolved meta-inflammation occurring in an obese state serves to illustrate how chronic, low-grade inflammation overlaps with metabolism to operate as a risk factor for a dampened anti-tumor immune response and increased risk of irAEs while simultaneously promoting improved outcomes with ICIs [185,186]. Accordingly, the association between excess weight in cancer patients receiving ICIs and significantly improved clinical outcome is especially marked when body mass index and irAEs are considered in combination [187,188]. If we aim to incorporate metabolic and dietary interventions to create more personalized cancer immunotherapy, it is necessary to take into account how important metabolic networks that sense and manage nutrients closely integrate with the activation of inflammatory pathways in immune cells to influence the physiological and pathological metabolic states [189,190,191,192]. Thus, more mechanistic research is needed before immunotherapy-boosting metabolic and dietary approaches can be incorporated into clinical practice to maximize anti-cancer responses while minimizing anti-self-responses to ICIs.

Combinatorial metabolic and/or dietary approaches to lower the risk of compromising ICI efficacy while preventing exacerbation of irAEs should consider also that the local cross-talk between microbes and cancer cells at the tumor site can provoke cancer-related inflammatory processes via the activation of immune cells [193]. Indeed, cancer-associated microbiota’s ability to modulate antitumor immune activation and therapy response might involve the direct provision to tumor cells of nutrients derived from microbial sources not only at the primary tumor sites but also at metastatic localizations [194,195,196,197]. This scenario adds even more complexity to the systemic (e.g., insulin-lowering effects of the ketogenic diet) versus cell-autonomous effects (e.g., restriction of serine/glycine or methionine) of dietary interventions [198], especially when considering recent evidence showing that diet-microbe interactions can alter the host response to drugs without altering the drug or the host [199]. Nonetheless, the extent of inter-patient variability in normal metabolism further highlights the need for controlled studies with metabolic therapies and/or dietary interventions in varied populations before new developments in combination with ICIs could be carried out.

## 6. Conclusions

Immunotherapy has had unprecedented success in halting even advanced cancer, and prolonging life in patients with highly aggressive tumors such as melanoma and lung cancer. Certainly, the growing arsenal of drugs unleashing the body’s immune system against tumors has captured the attention of society. However, although the percentage of cancer patients eligible to receive ICIs has increased in recent years to more than 40%, less than 15% of patients will respond to these treatments. The population-level effect of ICIs may be more limited, as only one in eight cancer patients would actually be expected to obtain a benefit from immunotherapeutics, even assuming that all eligible patients are treated. Therefore, although a greater percentage of cancer patients are becoming eligible for immune checkpoint blockade, the ratio of those benefiting is decreasing. These data reinforce the importance of identifying which patients are likely to benefit from immunotherapy, and which are going to be treated without any demonstrable benefit. This is critical, as these medications are exceedingly expensive, and so we need to rationalize which patients are most likely to benefit. Many scientists are engaged in tackling these crucial issues for the future of cancer immunotherapy by developing and testing a variety of biomarkers mostly involving tumor genomics, host germline genetics, and immune checkpoint levels. However, it is noteworthy that only half of all the variation in therapeutic efficacy of ICIs might be explained in terms of some of the most widely accepted determinants of immunotherapy response including the tumor mutational burden. A new wave of research into the molecular mechanisms of tumor-intrinsic resistance to immune checkpoint blockade has shed new light on the immunological implications of biological processes and signaling pathways such as interferon signaling, antigen presentation, WNT-β-catenin, cell cycle regulation, MAPK activation, and PTEN loss [167,168,169]. Here, we propose that the intercrossing of immune evasion and metabolic reprogramming cancer hallmarks at the tumor cell-intrinsic level should be viewed as a crucial dimension of cancer immunotherapy resistance that might guide the incorporation of new anti-metabolic drugs and dietary approaches to overcome primary, adaptive, and acquired resistance to ICIs. A deeper mechanistic understanding using clinical cancer metabolomics with stable isotope tracers has yet to be fully developed before tumor cell-intrinsic metabolic and dietary regulation of the immune checkpoint landscape could be successfully incorporated as a novel strategy capable of broadening the spectrum of cancer patients and indications that can be effectively treated with ICI-based cancer immunotherapy.

## Figures and Tables

**Figure 1 cancers-12-01757-f001:**
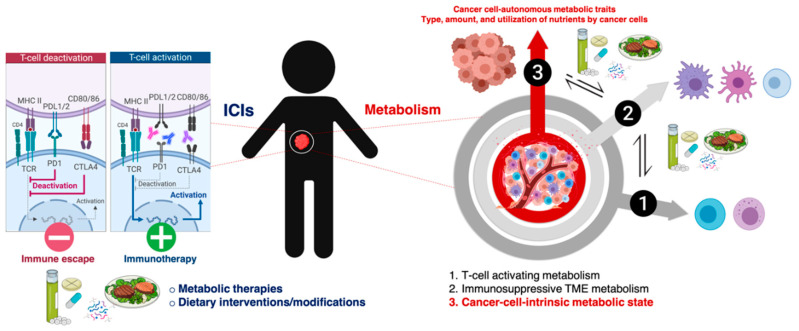
A three-layer metabolic regulation of the interface between tumor immune evasion and T-cell-directed immune checkpoint inhibitors. Layer 1: Metabolic specialization of T-cells. The activation, proliferation, and differentiation of T-cells are governed by dynamic changes in metabolic pathways involving glucose uptake/metabolism, mitochondrial function, amino acid uptake, and cholesterol/lipid synthesis. Layer 2: Immunosuppressive tumor microenvironment (TME) metabolism. Cancer cells compete with effector T-cells in the TME for the same pool of nutrients, leading to the strong inhibition of T-cells; the immunosuppressive activity of certain immune compartments (e.g., M2-polarized tumor-associated macrophages (TAM), myeloid-derived suppressor cells (MDSC), and regulatory T (T_reg_) cells) is highly dependent on the metabolic switches occurring in the TME. Layer 3: Cancer cell metabolic state. Certain metabolic nodes might suffice to drive the composition of the immune checkpoint landscape in cancer cells and the responsiveness of cancer cells to immunotherapy; diet-mediated changes in nutrient availability/utilization by cancer cells could have a similar impact on both the composition and/or functionality of the immune-escape mechanisms and the efficacy of immune checkpoint inhibitors. The amount, type, and utilization of nutrients derived from the diet as well as diet-altered activity of gut and tumor microbiota may influence not only the cell-autonomous metabolic capacity of cancer cells to shape their immune checkpoint landscape, but also the immunosuppressive metabolic features of the TME, and the metabolic specialization of T-cells. The pharmacological and dietary modulation of metabolic pathways, metabolites, and/or metabolic enzymes responsible for the antagonistic (e.g., tumor cells versus effector/cytotoxic T-cells) and symbiotic (e.g., tumor cells, TAM, MDSC, and T_reg_ cells) interplays of the TME and the microbiota might elevate the response rates of cancer immunotherapy. Balancing the inhibition of layers 2 and 3 while boosting layer 1 remains challenging without the delineation of the metabolic and dietary blueprints that shape the immune checkpoint landscape of tumor cells.

**Figure 2 cancers-12-01757-f002:**
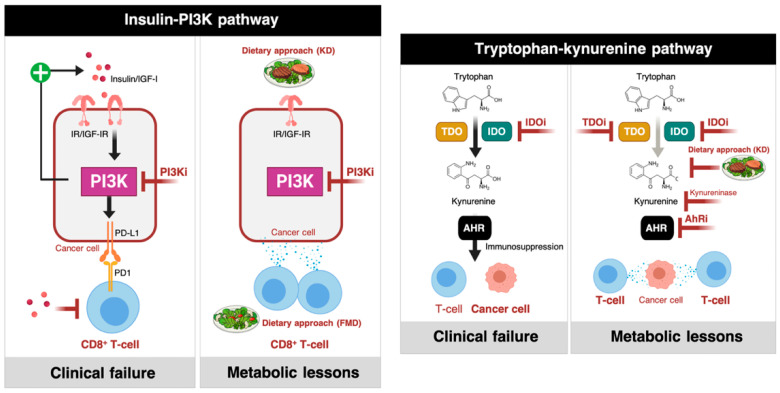
Metabolic flexibility and immune checkpoint inhibitors: learning from failures. The metabolic traits underlying the clinical failure of phosphoinositide 3-kinase (PI3K) (**left**) and indolamine 2,3-dioxygenase 1(IDO1) inhibitors (**right**) provide two excellent examples of how an in-depth understanding of the metabolic plasticity of tumors [64,65] can inform the use of metabolic and dietary interventions for more efficient ICI-based cancer immunotherapy.

**Figure 3 cancers-12-01757-f003:**
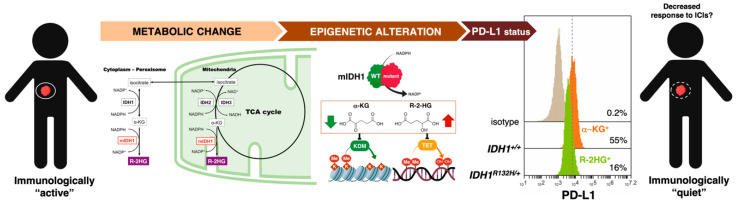
Oncometabolites can suffice to epigenetically regulate programmed death ligand 1 (PD-L1) expression in cancer cells. Beyond the well-recognized genetically-directed adaptations in nutrient acquisition (e.g., uptake of glucose and amino acids) and reprogramming of intracellular metabolic pathways (e.g., use of glycolysis/tricarboxylic acid cycle (TCA) intermediates for accelerated biosynthesis and NADPH production, increased demand for nitrogen, etc.), select metabolic activities and metabolites can directly affect the behavior and function not only of non-tumor cells residing in the TME, but also of cancer cells themselves via modification of the epigenetic landscape. Oncometabolites such as R-2-hydroxyglutarate (R-2HG), succinate, and fumarate are prototypes of such a class of cancer-promoting metabolites sharing a common causal mechanism in malignant transformation—namely the promotion of histone and DNA hypermethylation. Such rewiring of the epigenome causally drives the accumulation of undifferentiated cells with tumor-initiating capacity that might be accompanied by changes in the expression of immune checkpoints such as PD-L1. The ability of oncometabolites to drive immune quiescence in the TME while promoting DNA methylation in the regulation of immune checkpoint genes (*PD-L1*) might be of significance for the potential therapeutic application of immunotherapies in certain cancer subtypes (e.g., low-grade gliomas). However, it is also possible that the presence of abnormal proteins in isocitrate dehydrogenase 1 (IDH1)-mutated cancer cells may boost immune responses in other cancers. The figure shows that the introduction of the 2HG-producing mutant IDH1 enzyme in an otherwise isogenic background suffices to downregulate the expression of PD-L1 in MCF10A breast epithelial cells, a regulatory effect that can be specifically reverted by R-2HG-inhibiting and hypomethylating drugs [113].

**Figure 4 cancers-12-01757-f004:**
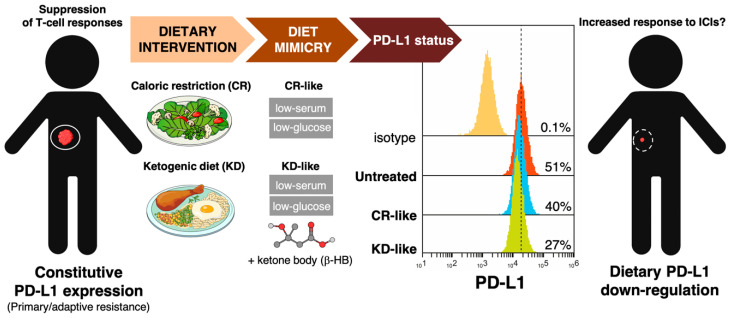
Ketogenic diet-mimicking culture conditions downregulate constitutive PD-L1 overexpression in biologically aggressive breast cancer cells. Tumor cells constitutively overexpressing immunosuppressive cell surface ligands such as PD-L1 may actively suppress anti-tumor T-cell functionality. To evaluate the impact of short-term fasting/ketogenic diet (KD)-like metabolic conditions on the constitutive expression of PD-L1, we used the basal-like/HER2-positive breast cancer cell model JIMT-1 which overexpress PD-L1 in 100% of the cells [122]. Although whether such a constitutive PD-L1 expression results in decreased or increased likelihood of responding to programmed death-1 (PD-1)/PD-L1-targeted immune checkpoint inhibitors remains to be determined, PD-L1-overexpressing JIMT-1 cells are intrinsically unresponsive to T-cells (unpublished observations). To mimic the physiological state of metabolites found in the circulation of patients treated with KD-like dietary interventions, we performed experiments in the presence of 2 mmol/L of the ketone body β-hydroxybutyrate (β-HB)—representing β-HB plasma concentration levels achieved under a KD or short-term fasting conditions (range 2–6 mmol/L)—and low (2.5 mmol/L) glucose levels to avoid an undesirable pathophysiological ketoacidosis. Such KD-mimicking physiological conditions reduced the expression of cell-surface-associated PD-L1 by approximately half.

**Figure 5 cancers-12-01757-f005:**
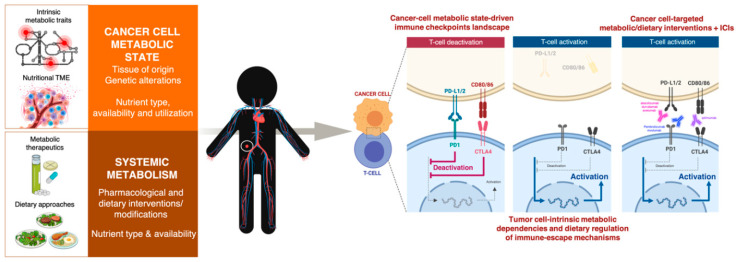
Metabolic state- and systemic metabolism-driven immune evasion landscape in cancer cells. Cancer cell metabolic states (which are determined by the tumor genetics, tissue of origin, and the cancer cell utilization of the available nutrients at the tumor microenvironment (TME)) and changes in the systemic metabolism of the host (which can be determined by the response to metabolic therapeutics and dietary interventions/modifications) can both alter cancer cell-intrinsic immune-evasion mechanisms, including the composition of the immune checkpoint landscape, and consequently dictate the responsiveness of tumor cells to immune checkpoint-based cancer immunotherapies.

**Figure 6 cancers-12-01757-f006:**
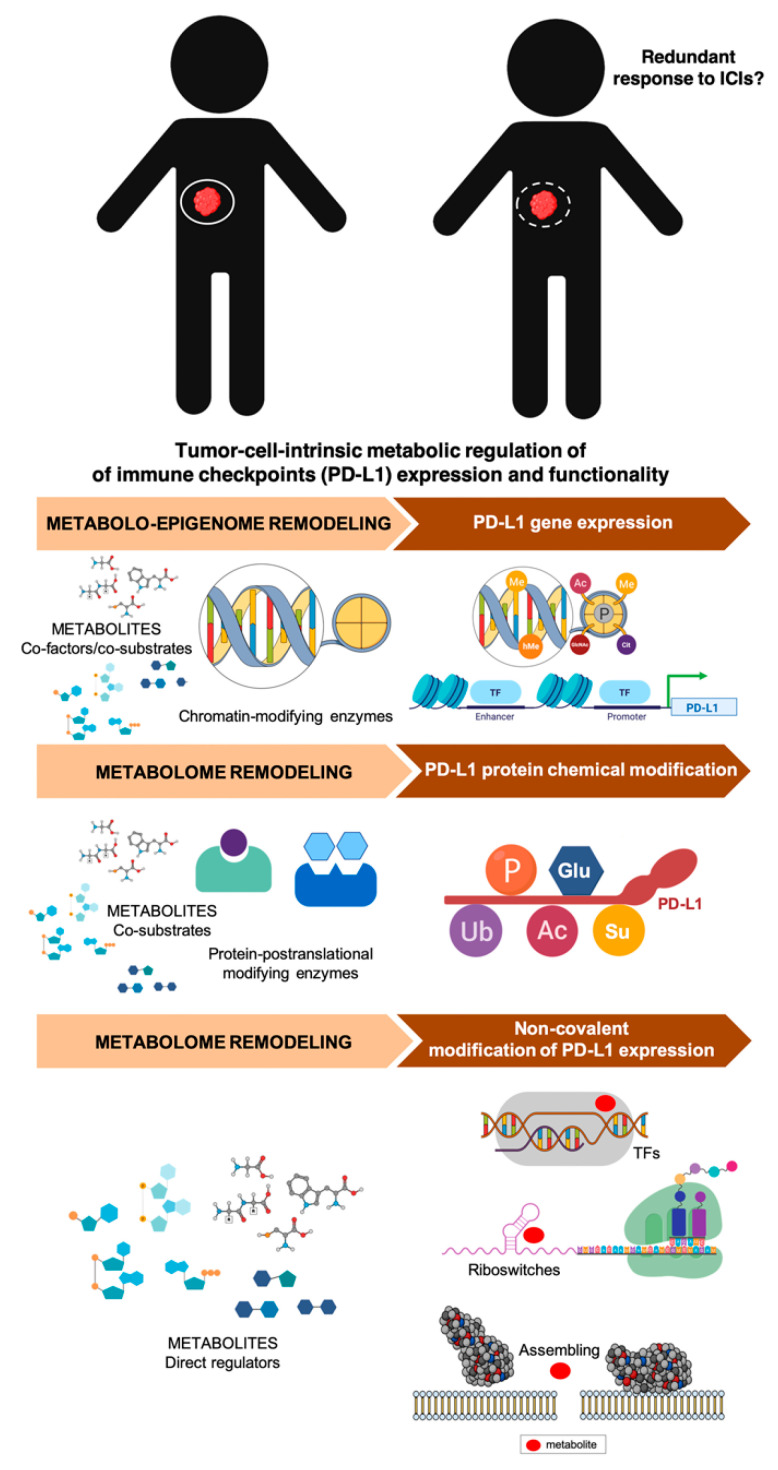
Metabolic mechanisms capable of regulating PD-L1 expression in cancer cells. PD-L1 expression can be modulated through various mechanisms including epigenetic and transcriptional regulation, post-translational, and non-covalent modifications. The metabolic regulation of these processes can limit tumor-specific PD-L1 expression and functionality, which might render PD-1/PD-L1 targeting with immune checkpoint blockade redundant.

**Figure 7 cancers-12-01757-f007:**
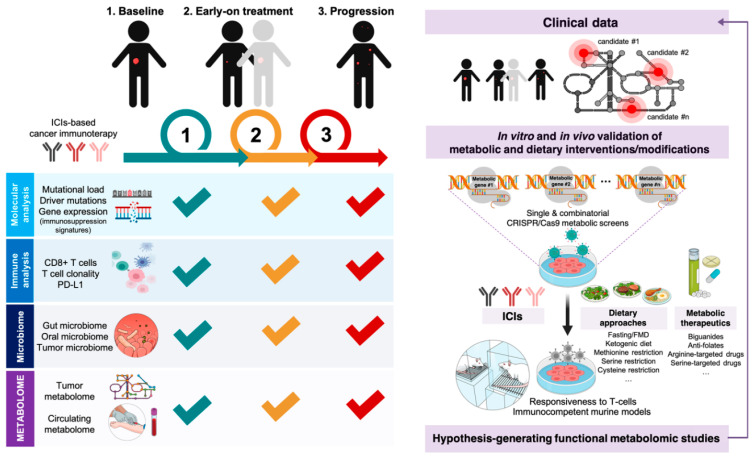
Monitoring metabolic mechanisms of resistance to immune checkpoint blockade therapy. The evaluation of longitudinal tissue and serum metabolomics at pretreatment, early-treatment, and progression time points and its integration with molecular, immune, and microbiome analyses might unveil potential metabolic mechanisms of therapeutic resistance to immune checkpoint blockade therapy. In vitro and in vivo validation of emerging clinical data and de novo discovery of cancer cells’ dependency on metabolic genes/networks that dictate responsiveness and resistance to immune checkpoint inhibitors using clustered regularly interspaced short palindromic repeats (CRISPR)/CRISPR-associated protein 9 (Cas9) screen-based screenings, metabolism-targeting therapeutics, and/or precision nutrition approaches, could provide a strong rationale for metabolically overcoming resistance to cancer immunotherapy.

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
