# Peer review of "Tumor Cell-Intrinsic Immunometabolism and Precision Nutrition in Cancer Immunotherapy"

_cancers, 2020, doi:10.3390/cancers12071757_

Round 1

Reviewer 1 Report

Cuyas et al. propose a very informative, substantial article on the interconnections between immune checkpoints and cancer immunometabolism. This is clearly at the forefront of the cancer research field. In this connection they discuss precision nutrition as a potential way of improving response to checkpoint blockade.

The review article is generally interesting but i have some general and some specific comments for improvement.

General:

1) The concept and description of precision nutrition does not cut through. What is proposed is a limited number of alterations to the diet that may aid in the ICT. However, a major component of how any diet will interface with the tumour and the tumour response to ICT, i.e. the gut microbiome, is overlooked. But it is very likely that the same diet will talk to the cancer in completely different ways in different individuals due to differences in the microbiome. Hence, I find it quite difficult, in humans, to make some of the generalisations that are made here. Discussion on the the interaction between nutrients and tumour through the microbiome filter  should be made and some of the precision nutrition discussion amended accordingly, with some of its conclusions toned down.

2) Some metabolites and/or pathways discussed are immunesuppressive in cancer but induce an opposite immune-response in other conditions. Discussions should be made that ICTs and targeting of metabolic pathways in cancer, while having positive effects for cancer therapy, could make patients suffer from autoimmunity and inflammatory diseases. As well as the dualism of the effect of certain metabolites and enzymes in cancer versus inflammatory conditions is worth mentioning.

3) The writing style, although grammatically correct, is not easy and does not help the reader. Most sentences are quite long and require re-reading to grasp the concept being discussed. I would suggest a revision of the style throughout to make the reading more accessible.

Specific:

1) In the Intro, the sentence at lines 56-58 "On the one hand..." is obscure.

2) In section 2, lines 146-153, please unlock this message and make it easier for the reader. is it because obese individuals have a potential for heightened inflammatory/cytotoxic responses that ICTs may work better?

Reviewer 2 Report

The authors of the perspective - "Tumor-Cell-Intrinsic Immunometabolism and  Precision Nutrition in Cancer Immunotherapy" did a good job in reviewing and compiling the literature of Cancer Immunotherapy from last few years. However, the manuscript figures need major improvements so that reader would find it easier to follow the review.
Figure 1 needs to be improved significantly - (i) It is impossible to read and understand the T-cell activation/deactivation pathway from the figures. The legends and the descriptions are too small to be comprehended on a regular computer screen without 400% magnification. (ii) All other texts on the right side of the figure need to have a higher size font.
I have similar problems with Figures 2, 4, 6, and Figure 8 - the reader will have a tough time understanding the message as they are difficult to read. All these figures need to be redrawn with larger font sizes.

Round 2

Reviewer 1 Report

The manuscript is acceptable for publication.

Reviewer 2 Report

I am satisfied with the improvements.